# Modulated Fluorescence in LB Films Based on DADQs—A Potential Sensing Surface?

**DOI:** 10.3390/molecules27123893

**Published:** 2022-06-17

**Authors:** Marek Szablewski, Richard L. Thompson, Lars-Olof Pålsson

**Affiliations:** 1Department of Physics, Durham University, Durham DH1 3LE, UK; marek.szablewski@durham.ac.uk; 2Department of Chemistry, Durham University, Durham DH1 3LE, UK; richard.l.thompsonl@durham.ac.uk

**Keywords:** DADQ, Langmuir-Blodgett film, fluorescence, sensing

## Abstract

Novel fluorescent Langmuir-Blodgett (LB) films have been constructed from three different amphiphilic dicynaoquinodimethanes (DADQs). The DADQs varied in functional group structure, which had an impact on the LB film structure and the fluorescence properties. As the fluorescence of DADQs competes with non-radiative decay (conformational change), the packing and/or free volume in the LB film will influence the average fluorescence lifetime and integrated intensity. The pristine (blank) LB films were then exposed to a selection of non-fluorescent target analytes (some with environmental relevance) and the fluorescence was measured and analyzed relative to the pristine LB film. Exposure of the LB films to selected target analytes results in a modulation of the fluorescence, both with respect to average fluorescence lifetime and integrated intensity. The modulation of the fluorescence is different for different DADQ LB films and can be attributed to restricted non-radiative decays or charge transfer reactions between target analyte and DADQ LB film. The response from the DADQ LB films shows that these systems can be developed into sensing surfaces based on fluorescence measurements.

## 1. Introduction

Langmuir-Blodgett (LB) films consist of highly ordered nano structures of mono or multi layered amphiphilic molecules. LB films can be engineered for numerous applications including functional coatings, sensing surfaces, etc. [1,2,3,4,5,6,7,8,9,10,11], but LB films also resemble naturally occurring biological cell membranes. Despite widespread applications, there are, to our knowledge, very few examples of fluorescent LB films. Some examples from the literature include studies from the authors in [12,13,14,15,16,17,18]. Such systems would be very attractive in, for instance, sensing applications, due to the inherent sensitivity of the electronic excited state and the modulation of the fluorescence, or, in more general terms, the photoluminescence.

A major obstacle in the pursuit of fluorescent LB films is the availability of amphiphilic organic materials with intense fluorescence in the optical visible part of the electromagnetic spectrum, i.e., excitation and emission at wavelengths in the range 400–700 nm. We have elected to work with adducts from 7, 7, 8, 8-1,4-[(NC)_2_C]-C_6_H_4_ or tetracyanoquininedimethane (TCNQ), which can readily be synthetically modified to form dicynaoquinomethanes (DADQs) with generic formulae’s 4-(R_1_,R_2_C)-1-[-[(NC)_2_C]-C_6_H_4_, where R_i_ denotes the functional group [19,20,21]. Indeed, DADQs are a unique class of materials that meet the criteria of amphiphilic character and possess the optoelectronic properties required to produce intense and environmentally dependent fluorescence [22].

Initially, there was considerable interest in the non-linear optical (NLO) properties of DADQs driven by a large figure-of-merit; hyperpolarizability (*β*) in combination with a large ground state dipole moment (${\bar{\mu }}_{g}$). This figure-of-merit, ${\bar{\mu }}_{g}\times \beta$, is relatively large for this class of organic materials, and applications in second harmonic generation (SHG) was indeed envisaged [21,23,24,25]. However, the occurrence of fluorescence (although weak) complicated the use of DADQs in NLO. It was immediately realized that the fluorescence from DADQs displays some very unusual features [22]. A very strong dependence on the dielectric and viscoelastic properties of the medium is generally observed and the origin of this can be traced to the optoelectronic properties of DADQs. Ground state dipole moments in DADQs have been found to be very large: ${\bar{\mu }}_{g}=15-20$ D, verified both experimentally and theoretically [22]. Upon optical excitation, the excited state dipole moment is found to be significantly smaller, ${\bar{\mu }}_{e}\leq 1.0$ D, and this leads to a highly unusual negative solvatochromism [22,26].

Perhaps more significant is the impact of the medium viscoelastic properties on the fluorescence of DADQs. The ground state conformation of DADQs is a non-planar configuration of molecular planes with a dihedral angle in the range 20–45°, dependent on functional group (R_1_ and/or R_2_) structure [22,26]. Optical excitations disrupt this twisted ground state configuration, and the molecular planes tend to planarize, resulting in zero dihedral angle between functional group structures and the benzenoid ring plane. However, it should also be noted that the dihedral angle between the benzenoid ring structure and the CN groups can increase upon optical excitation [27]. The viscosity of the medium leads to shear friction between solvent and solute (DADQ), which hinders the planarization. As conformational change in the dihedral angle is a non-radiative excited state process, there will be a competition between this process and that of the radiative decay, which is the fluorescence. The competition between these processes can be monitored via the fluorescence quantum yield:(1)$$\phi_{f}=\frac{k_{R}}{k_{R}+\sum k_{\mathrm{nr}}}=\frac{\tau_{f}}{\tau_{0}}$$
where $k_{R}$ is the radiative decay rate and $\sum k_{\mathrm{nr}}$ is the sum of all non-radiative decay processes including the planarization of molecular planes, $\tau_{f}$ and $\tau_{0}$ are the experimentally measured fluorescence and natural lifetimes, respectively. Accordingly, in low viscosity media the fluorescence intensity is low (fluorescence quantum yields in the order of 10^−2^–10^−3^) with a short fluorescence lifetime $\tau_{f}$, dominated by a large $\sum k_{\mathrm{nr}}$ contribution [22,26]. By contrast, in high viscosity media, the fluorescence is more intense ($\phi_{f}$ ≈ 0.2) and measured fluorescence lifetimes are generally longer due to less $\sum k_{\mathrm{nr}}$ contributions [22]. The same trends are also observed for more extensive functional group structures (R_1_ and/or R_2_), which leads to more shear friction and more intense fluorescence/longer fluorescence lifetimes.

It has been shown in earlier works from our laboratory that DADQs can be used for the construction of LB films [23,24]; this was exploited in devices for SHG, thereby achieving non-centrosymmetric systems. However, the fluorescence properties of these LB films were never investigated. In this work, we show that intense fluorescence can be produced in LB films based on amphiphilic DADQs (see Figure 1 for structures). We also demonstrate that exposure of the surface to selected (non-emissive) target analytes result in a modulation of the fluorescence. The target analytes chosen to test the response of DADQ LB film surfaces all have environmental relevance and/or physiological relevance. The target analytes chosen in this work were NiO, bisphenol A (Bis A), humic acid (HA) and human serum albumin (HSA). Of the ones chosen here, some can be present in water streams, e.g., Bis A and HA, and could have a serious impact on the health and viability of many bio and ecological systems. HSA was selected as it is a (relatively) large molecular weight (biological) macro molecule.

A modulation of the fluorescence was indeed observed and these fluorescent DADQ LB films could therefore, in principle, be developed into a novel sensing platform. This type of sensor could be used in a range of applications, including medicine and/or environmental science.

## 2. Results

### 2.1. LB Film Isotherms and LB Film Deposition

Isotherms of DADQs **1**, **2** and **3** are shown in Figure 2. Table 1 shows the data related to isotherms and the LB film depositions onto pre-treated glass slides. DADQs were deposited in Z-type deposition with the functional group structures exposed to the air interface, thereby fully exploiting the variation in functional group structure for molecular interactions to target analytes.

Molecular head areas were determined by a fitting procedure using linear regression on a subset of the data, from what was determined to be the solid phase (just before the collapse of the film). Extrapolating the fitted line to the “Molecular Area” axis (see Figure 2) provides the molecular head area for each DADQ.

The molecular head area a was then calculated from:(2)$$a=\frac{A}{cN_{A}V}$$
where *A* is the trough area (from the fitting procedure), *c* is the concentration, *N_A_* Avogadro’s number and *V* is volume of material deposited.

As expected, **3** has the largest molecular head area, which can be rationalized by its more extended functionalized group structure. The molecular head area of **1** is larger than that of **2,** which can be explained by the presence of the –NH group on the ring structure, in combination with a slightly larger ring structure compared to **2** (six-membered ring structure instead five in **2**). The uncertainties in the molecular head areas have a few different sources, including the standard error from the linear regression fit in combination with uncertainties in concentration and deposited volume. From the isotherms, conditions for LB film deposition onto solid platforms were determined. A key parameter in this context is the target surface pressure, which is given for each DADQ in Table 1. These values were chosen on the basis of the observed surface pressure at LB film collapse; each target surface pressure was accordingly set to be slightly below the collapse pressure.

The transfer ratio (the ratio between the decrease in the monolayer area during a deposition stroke, A_l_, and the area of the substrate, A_s_) values are a good indicator of the success of film deposition. The average transfer ratio was calculated from that of the 2-layer LB film (unless otherwise stated). The ratio should, in theory, have a maximum value of 1.0; however, sources of error could arise from the design of the NIMA trough that causes the molecules to be deposited onto the structure of the dipping mechanism. Furthermore, subtle vibrations in the environment can lead to monolayer collapse. The transfer ratio of **2** is significantly larger than 1.0, as seen in Table 1. A possible explanation for this could be that some material may have gone into the bulk and therefore neither deposited nor left on the water surface. This could be caused by the lower amphiphilic character of **2** compared to **1** and **3**. This could also have an impact on the lower collapse pressure of **2** compared to **1** and **3**.

It is informative to examine the compressibility of the LB films, as this provides information about film structure and packing, a complementary measure to the molecular head area. The compressibility *C* can be calculated from:(3)$$C=-\frac{1}{a}{\left(\frac{\partial a}{\partial \pi }\right)}_{T,p,n}$$
where *a* is the molecular head area, *π* is the surface pressure, and the partial derivate term is the slope of the line extrapolated and fitted from the solid phase. The calculated compressibility of these is listed in Table 1, with LB film **2** returning the largest value, followed by LB films **3** and **1**.

### 2.2. Atomic Force Microscopy

To obtain a more detailed picture of the LB film surface structure, **1** was investigated by means of atomic force microscopy (AFM). This was chosen as it is reasonable to assume that this DADQ molecule would form the most even LB film of the three studied here; **1** has the least extended functional group structure. The AFM height map shows the surface of the multilayer deposition of **1** for a 4-layer LB film. This is a typical height map of LB films, as shown by the several spikes on the surface and randomly distributed areas of larger thickness. AFM is a sensitive technique that is easily affected by the crevice on the surface, which may entrap the sample on the glass surface as it is being probed.

Images from the AFM microscope confirm that the surface has some roughness due to the underlying structure of the glass slides. We also note that the height scale in Figure 3 is up to 22 nm, but this has been scaled by the spikes present on the surface. It is difficult to obtain the LB film thickness of the 4-layer film of **1** in Figure 3, but an estimate based on the C–C bond length suggested a height of 1.5–2.0 nm for 4 layers.

Exposing the LB film surface of **1** should, in principle, lead to a changed AFM height map. In Figure 4, a high molecular weight biological system HSA has been exposed to the LB film surface. This leads to a very different surface height profile, as can be seen in image B (Figure 4B). Many proteins have a relatively loose structure and, as a consequence, they will spread out and smoothly cover the surface they interact with. Before the exposed films were examined with AFM, we allowed the solvent (water) dry off under normal ambient conditions. The image (Figure 4B) from the HSA exposure is therefore likely an aggregation of protein on the LB film surface.

Exposing the LB film surface of **1** with a lower molecular weight organic complex, HA, led to a less pronounced impact on the surface height profile, as observed in AFM (Figure 4A). HAs are not well defined or structurally unique organic matter. Instead, they are mixtures with variations of catechol, quinone, phenol and sugar moieties. The molecular weight of HAs is therefore not well defined. The height profile of the surface is, in the case of HA exposure, only ~10 nm above the pristine surface shown in Figure 3. This is in sharp contrast to the (almost) μm height of the surface exposed to HSA. A cross-sectional view of the deposition of HA is shown in the SI (Appendix A) and this shows a height profile of a HA droplet consistent with this height. It is therefore clear that HAs, although complex and of some moderate size, only have a minor impact on the surface profile of the LB film of **1**.

From the AFM height map in Figure 4A, we estimated the area covered by HA deposition that could be tracked in AFM. The areas of interest were the ones with the highest height profile (in the 30–34 nm range, indicated in yellow in the AFM map). The rationale for this was to explore one aspect of the detection limit for fluorescence modulation in the LB film. In this estimate, approximately 10% of the area in Figure 4A appears to be covered by HA. Interestingly, with a deposition of NiO and Bis A, we could not detect and change to the height map in the AFM experiment. With the underlying roughness of the glass slide, it is very likely that the small height difference for these target analytes can escape detection.

### 2.3. Time-Resolved Fluorescence Microscopy

Fluorescence microscopy has the advantage of high sensitivity and the ability to probe local μm size areas of the deposited LB films, with and without exposure to target analytes. Particularly, time-resolved fluorescence microscopy was employed, as it provides information not only on the fluorescence kinetics but also on relative intensities between pristine (blank) and exposed LB films. In general, we observed triple-exponential decay of the fluorescence with a first component $\tau_{1}\approx 30-300$ ps, $\tau_{2}=500-700$ ps and a third component $\tau_{3}=1.0-5.1$ ns. No fit of the data was accepted unless the $\chi^{2}\leq 1.5$. Figure 5 displays the fluorescence decay of pristine (blank) LB films of **1 2** and **3** for 2-layer deposition. The origin of the first component $\tau_{1}$ is somewhat uncertain, as it could be due scattered light rather than fluorescence from the LB films, or a combination of both. The IRF has a temporal width in the order of ~100 ps, which is close to the fitted value of $\tau_{1}$ and is therefore close to the limit of the time-resolution of the detection system. However, it is also important to point out that for pristine (blank) LB films, this decay phase does not dominate the overall decay of fluorescence.

The deposition of target analytes could, in principle, lead to interactions between the electron accepting functional group structure of the DADQs and the target analyte. This results in the modulation of the fluorescence lifetimes and the integrated intensity. For deposition with organic materials on the LB film of **1**, the relative amplitude (and yield) of the third component $A_{3}\left(\lambda \right)$ is affected with an increase in amplitude (and yield) as a consequence. For LB films of **2** and **3,** the outcome target analyte deposition is more varied. To demonstrate more clearly the impact on the fluorescence kinetics due to target analyte deposition, we elected to monitor the average fluorescence lifetime. The average fluorescence lifetime is given as [28]:(4)$$\langle \tau \rangle =\frac{\sum A_{i}\left(\lambda \right)\cdot \tau_{i}^{2}}{\sum A_{i}\left(\lambda \right)\cdot \tau_{i}}$$

Time resolved fluorescence data acquired from pristine LB films show the differences in fluorescence decay between LB films from the three different DADQs (see Figure 6 for data). Regarding fluorescence lifetime, **2** produces the longest average, with **1** being intermediate and **3** the shortest.

A different approach to the time-resolved fluorescence lifetime analysis is to examine the intensities via the integration of the fluorescence decay obtained in the fluorescence microscopy experiments. This can be conducted at an acceptable precision as we can control sample geometry and fluorescence excitation and detection conditions reliably in the microscope. This allows a quantitative comparison to be made between different DADQ LB films, with and without target analyte exposure. Ideally, a fluorescence quantum yield measurement would be preferred, but the extremely low absorption cross-section of a two-layer LB film makes this a very difficult experiment, with many uncertainties as the only clear outcome. Furthermore, the detection of the fluorescence in the microscope is of superior sensitivity compared to the fluorescence spectrometer, where factors such as sample geometry and emission collection efficiency can introduce systematic errors. However, intensity normalized steady state fluorescence spectra from LB films of **1**, **2** and **3** can be found in the SI (Appendix A), showing fluorescence maximum at ~480 nm, ~450 nm and ~525 nm, for **1**, **2** and **3**, respectively. The longer wavelength of fluorescence for **3** is consistent with its more conjugated structure compared to the other two DADQ systems.

The accumulated intensity from the time-resolved fluorescence microscopy was simply obtained through the numerical integration of the decay F(λ,t) as:(5)$$Int=\int_{0}^{\infty }F\left(\lambda ,t\right)\cdot dt$$

The relative intensities of 2-layer pristine LB films from **1**, **2** and **3** are shown in Figure 7. Several spots on the LB film were probed to obtain data, as seen in Figure 7, with its associated uncertainty. Here, we have chosen **1** as our reference system and normalized the intensity to 1.0. The intensities of **2** and **3** are therefore scaled accordingly. The uncertainties are large for the integrated intensities, as shown in Figure 7, indicated by the error bars. We attributed this to the variations in film quality across the slide. A picture of a full-size slide of pristine LB film of **3** is shown in the SI (Appendix A), in which variations on a macroscopic scale, partly due to the glass slide platform, are clearly seen. We remark that the photon counting module used in the time-resolved fluorescence microscope has some wavelength dependence on the detection efficiency. However, in this wavelength range (480–550 nm), it is reasonably flat and the detector response is within the error bars for pristine films in Figure 7.

While the relative intensities of **1** and **2** are congruent with the average lifetime data, **3** shows a deviation as the intensity is intermediate between **1** and **3**, while the average fluorescence lifetime is the shortest measured here.

For target analyte exposure, we find that in some material-LB film combinations, no significant impact is observed, suggesting that interactions are weak and/or insignificant. Figure 6 shows the complete set of average fluorescence lifetime data for all material combinations. Target analyte exposure was carried out through the application of liquid droplets (letting the LB film dry before measurements). The linear response on the fluorescence with target analyte exposure was not considered in this work. Instead, we focused on the chemical composition and size of target analytes. With μL droplet volumes and mM target analyte concentrations; 10^−8^–10^−9^ moles of target analyte covering an area of μm dimensions, modulation of the fluorescence can be readily detected using fluorescence microscopy.

We are confident that the intactness of the LB film structure is not affected in this procedure. Controls were carried out using long chain alcohol (pentanol), a solvent that can dissolve the LB films, although the dissolving power is low (compared to, e.g., methanol or ethanol). The time-resolved fluorescence microscopy then shows that decay is dominated by a fast decay phase in the range 25–70 ps, with considerable amplitude and yield (temporally limited by the IRF of the detection system), as seen in the SI for LB films from **1** and **2** (Appendix A). The explanation for the appearance of this fast decay phase is that DADQ adducts have been detached from the surface and are in effect in a solution phase. The fast decay phase is therefore a reflection of the conformational change in the DADQ adduct in the solution [26].

For LB film **3,** the scenario is slightly different, as a fast decay phase of 30–70 ps is always present in the overall decays for the fluorescence, albeit with variations in amplitude. Pentanol deposition does not markedly change the fluorescence decay, but this decay phase is also present in the blank (pristine) film and does not change amplitude and yield significantly with depositions (Appendix A). We are also confident here that exposure by the selected target analytes does not dissolve the LB film of **3**.

For NiO, we observed that for all three different LB films, deposition does not result in a metastable state in the μs–ms range. This indicates that the rate of intersystem crossing into triplet states must be very small for all DADQ LB films. We also observed that for the deposition of HA, a reduction in the intensity is observed for all three DADQ LB films. However, the average lifetime is prolonged for **1** and **3,** which is inconsistent with an intensity decrease. It is likely that HA oxidized the LB film surface, particularly for LB film **2**, which would explain the low intensities obtained. However, we are confident that oxidation of the LB film does not occur during the deposition of the other target analytes. The increase in integrated intensity observed in some of the material combinations would be inconsistent with such an effect. An alternative explanation is fluorescence quenching due to charge transfer, which could be caused by a hydrogen bond interaction between LB film surface and the target analyte.

## 3. Discussion

Time-resolved optical spectroscopy (using fluorescence and/or transient absorption) has previously shown that the kinetics of DADQs in low viscosity media is relatively fast (rates ~10^12^ s^−1^), which can be rationalized by rapid and unrestricted excited state conformational change (planarization of molecular planes) [22]. In LB films, the free volume is limited and tight packing, and the orientation of DADQ molecules will restrict conformational change. This has the consequence of slowing down the excited state fluorescence kinetics, but with the benefit that fluorescence quantum yield and intensity increases (smaller $\sum k_{\mathrm{nr}}$ term in Equation (1)). For DADQ LB films, three kinetic terms are generally required to satisfactorily fit the data. The faster of two components is interpreted as being a reflection of DADQs with some degree of excited state conformational change (but with the caveat that the origin $\tau_{1}$ can also be due to scattered light, as previously mentioned), while the slower third component represents the population of fixed DADQ molecules that are unable to planarize due to tight packing in the LB film. It is also noteworthy that the shape of the fluorescence spectra from the LB films (see SI, Appendix A) is distinct from the solution phase, as shown in previous work from our laboratory [22,26]. The fluorescence bands from the LB films are narrower, which is indicative of a smaller distribution of conformational sub-states. This is, in-turn, can be explained by packing and limited free volume in the LB films which will restrict rotation of dihedral planes. The isotherms all gave different molecular head areas for each DADQ, and the average lifetime data suggest that there is a correlation between the molecular head area for all three DADQs in pristine films. The smallest molecular head area was seen in **2**, which also displays the longest average fluorescence lifetime. This indicates that in the LB film of **2**, packing must be tight (little free volume) and, as a consequence, the conformational change is restricted (small $\sum k_{\mathrm{nr}}$ term). In this context, we also refer to the compressibility of the LB films, where the LB film of **2** shows the highest value, which is indicative of tight packing and little free volume. As the fluorescence lifetime is given by $\tau_{f}={\left(k_{R}+\sum k_{\mathrm{nr}}\right)}^{-1}$, smaller $\sum k_{\mathrm{nr}}$ will result in longer $\tau_{f}$. At the other extreme is **3**, which has the largest molecular head area in conjunction with the shortest average fluorescence lifetime. Free volume in the LB film of **3** must therefore be larger, leading to more conformational change and a more significant $\sum k_{\mathrm{nr}}$ term. The LB film of **1** fits in to this trend as it is an intermediate, both with respect to the molecular head area and average fluorescence lifetime. The compressibility is larger in the LB film for **3** (as compared to **1**), which goes against the trend of molecular head areas. It is noteworthy that molecular head areas are close; however, comparing LB films from **1** and **3** and considering the molecular head area uncertainties, it is difficult to completely distinguish **1** from **3**.

As stated earlier, the surface pressure for deposition was set to lower values than the measured collapse pressure. Balaswamy et al. [14] show that intermolecular interactions in LB films can be controlled through the deposition pressure. Control of these interactions is important in this context, as they have an impact on the optoelectronic properties of the fluorescent chromophores. Hestand and Spano [29] invoke a model of H-aggregates, which is relevant in the case of LB films, as the molecular orientation and relative positions could, in principle, lead to H-aggregation [29]. In this model, dipole–dipole interaction between a pair of interacting dipoles ${\bar{\mu }}_{1}$ and ${\bar{\mu }}_{2}$ is given as:(6)$$J=\frac{{\bar{\mu }}_{1}{\bar{\mu }}_{2}}{4\pi \varepsilon_{r}R^{3}}\left(1-\cos^{2}\theta \right)$$

In this formalism, the angle *θ* represents the angle between the midpoints of the interacting dipole moments for H-aggregates $\theta =0^{o}$ (for J-aggregates $\theta =45^{o}$; midpoint to end of adjacent dipole [29]). This was used to estimate the coupling in pairwise dipole–dipole interactions, as it is well known that the ground state dipole moment of DADQs is large, and this could, in turn, have an impact on the fluorescence, as previously stated. We assumed dipole moments ${\bar{\mu }}_{1}={\bar{\mu }}_{2}=20$ D, which are on the larger side but in line with many other DADQs [22,26]. We used a center-to-center distance of R=5 Å, which is reasonable given the size of the functional group structures of DADQs **1, 2** and **3**. The dielectric medium is not well-characterized in LB films but relative permittivity in the range $\varepsilon_{r}=3.2-6.3$ has been suggested by Pearson et al. [30]. Given these uncertainties, it is reasonable to give a range of interaction energies instead of a specific value and the estimated range is J=100−1000 cm^−1^. However, this can be compared to coupling energies in squaraine aggregates which are in the range 6000–12,000 cm^−1^, pending on molecular structure [29]. Hence, we would argue that interactions are weak in the LB film of this work, and thus not in the range when H-aggregation prevails and impacts on optoelectronic properties. This is also supported by the steady state optical spectroscopy on films; although of poor quality, it appeared that no spectral shifts consistent with H-aggregate formation in LB films were observed (see SI for fluorescence from LB films).

Exposing pristine LB films to target analytes modulates the fluorescence, both with respect to average fluorescence lifetime and intensity. In an ideal scenario, one would expect the average fluorescence lifetime and the intensity to vary in the same direction with target analyte exposure (relative to the pristine LB film), e.g., a longer fluorescence lifetime in combination with an enhanced intensity would imply an interaction between the target analyte and the LB film that would lock the functional group structure. This would, in turn, restrict the conformational change and therefore decrease non-radiative decay. The data show that LB films based on **1** respond in this way to target analyte exposure with organic materials. In the functional group structure of **1**, there is an –OH group which obviously could act as a donor in hydrogen bond interactions. Because of the tight packing in LB films of **2**, there is very likely not much room to further restrict conformational change. Indeed, exposing LB films of **2** with target analytes reduces the average fluorescence lifetime and intensity. In this case, it is more difficult to interpret the data, but a possible scenario is excited state quenching due to charge transfer processes.

The LB films of 3 show the most complicated modulation of the fluorescence with target analyte exposure. From HSA exposure, no significant modulation of the fluorescence is observed, suggesting that interactions between the LB film and HSA are weak. The functional group structure of 3 has more aliphatic character compared to 1 and 2, which in turn could reduce and/or repel interactions to polar materials or materials with charged surfaces such as HSA. There are many factors that are crucial in protein adsorption to surfaces and these include interfacial ion distribution, interfacial charge regulation of amino acids in the proximity of the surface, charge neutrality, and mass balance [31,32]. This would suggest that interactions between HSA and the LB film of 3 are possibly unfavorable.

An admission is that some of the observations as tabulated in Table 2 are difficult to rationalize, but we have elected to show all the data for completeness. In particular, for cases where the average fluorescence lifetime and the integrated intensity is modulated in opposite directions, we refer to these as anomalies. We can partly explain these with the relatively large uncertainties in the calculated integrated intensities. However, we remark in this context that an alternative, and possibly more informative, approach could be to consider the fluorescence lifetime variations only. This parameter is independent of LB film thickness and/or other impacts on the LB film such as oxidation or a dissolved LB film.

However, what is clear from the data is that modulation of the fluorescence can be observed for low molecular weight target analytes such as HA and Bis A. For this type of target analyte exposure, the AFM does not show any significant impact on the LB film surface profile (none observed for Bis A). Only for a large molecular weight system such as HSA can a changed surface profile be clearly observed. This is broadly support by other studies where AFM is used to study surface topology upon target analyte exposure [33,34,35]. This shows the superior sensitivity in fluorescence-based measurements.

Another aspect of sensitivity and detection limits is the coverage of material on the LB film surface required to see the modulation of the fluorescence. Clearly, molecular interactions are important in this context, as in some material combinations target analytes may not stick to the LB film surface (as alluded to earlier). For HA exposure to the LB film of 1, a clear modulation of the fluorescence could be observed, both in lifetime and intensity variation. From AFM, an HA area coverage of 10% was found (as discussed earlier), which clearly has some associated uncertainties. However, it is reasonable and safe to conclude that a surface area cover of >20% should result in a detectable modulation of the fluorescence.

## 4. Materials and Methods

### 4.1. Materials

Three different amphiphilic DADQs (with single R_1_ and R_2_ functional groups structures) for LB film fabrication were used in this study; **1** (2-(4-(5-(hydroxymethyl)-1,3-oxazinan-2-ylidene)cyclohexa-2,5dien-1-ylidene)malononitrile), **2** (2-(4-(4-butyloxazolidin-2-ylidene)cyclohexa-2,5-dien-1ylidene)malononitrile) and **3** (2-(4-((hexylamino)(3-hydroxypyrrolidin-1-yl)methylene)cyclohexa2,5-dien-1-ylidene)malononitrile). Synthetic procedures with characterization for **1**, **2** and **3** can be found in the SI section.

For exposure to the LB film surfaces, a few non-fluorescent target analytes were judiciously selected and used without further purification and/or manipulation; Human Serum Albumin (HSA) Sigma Aldrich, Humic Acid (HA) Sigma Aldrich, Nickle Oxide (NiO) Sigma Aldrich, Bisphenol A (Bis A) Sigma Aldrich. All the target analytes were dissolved in deionized water to < 1 mM concentrations and deposited in μL volume droplets to pristine LB film surfaces. The LB films were then left to dry before measurements.

### 4.2. LB Film Fabrication

A Langmuir-Blodgett Trough NIMA model 2022 was used for LB film deposition. Solutions of 0.1 mg/mL of DADQs **1**, **2** and **3** dissolved dichloromethanes (DCM) were prepared for LB film deposition. The surface of high purity water (18 MΩ resistivity) in the trough was thoroughly cleaned. Sufficient cleanliness of the water surface was achieved when the surface pressure reached −72 mN × m^−1^. Next, 400–800 µL of the DADQ solution was deposited onto the surface at a 45-degree angle with the tip of the Hamilton needle aimed as close to the surface as possible. Droplets were applied at even points across and left for DCM to evaporate for 20 min to form a 2D monolayer. The surface was annealed multiple times and an isotherm of surface pressure vs. molecular area was recorded to determine the quality of the 2D monolayer. The isotherms were further used to determine the surface pressure required to reach the solid phase. Pre-treated clean microscope glass slides were used for dipping and deposition of LB films to a solid (and transparent) platform. Target surface pressures were set for each individual DADQ amphiphile. Z–type deposition onto the glass slide was chosen to fully exploit the DADQ functional group structures (CN–groups attached to the solid platform with functional group structure pointing up away from the platform surface). LB films with 1 up to 4 layers were fabricated.

### 4.3. Time-Resolved Fluorescence Microscopy

Time-resolved fluorescence microscopy was performed as previously outlined in [36] but with some modifications. Briefly, for point scanning time-resolved fluorescence microscopy, a home-built system was used based on a Zeiss Axiovert 135M Inverted Epi-fluorescence microscope. The excitation source was a PicoQuant diode laser LDH-P-C-395 (395 nm, 70 ps pulse FWHM@20 MHz). A Zeiss 40×/LWD (long working distance) lens was used for the detection of the fluorescence. The fluorescence was detected using long-pass filters LP 420 or LP570 (Comar Instruments). The electronic detection system was based upon the well-known time-correlated single photon counting technique (TCSPC) with a photon counting module Idquantic (id100–20), in combination with a Becker-Hickl SPC-130 TCSPC module. The data were subsequently fitted to a sum of exponentials using the software FluorFit (www.fluortools.com (accessed on 9 April 2020)). The IRF in the time-resolved fluorescence microscopy was obtained through light scattering from Ludox particles dispersed on a microscope slide. The FWHM was ~90 ps, which, in principle, afforded a temporal time-resolution of ≥20 ps. The data were subsequently fitted to a sum of exponentials:(7)$$F\left(\lambda ,t\right)=\sum_{i}A_{i}\left(\lambda \right)\exp\left(-k_{i}t\right)$$
by deconvolution with the instrument response function (IRF). In Equation (7), $A_{i}\left(\lambda \right)$ is the amplitude (in principle wavelength dependent) and $k_{i}={\left(\tau_{i}\right)}^{-1}$ represents the fluorescence lifetime. Fitting the data over the whole IRF was problematic and did not result in acceptable overall fits of the data. This is likely due to the complicated geometry and the many surfaces in the microscope. We therefore reverted to a tail fitting procedure from the maximum intensity point in the decay. Quality of fit was assessed by the chi square and a visual inspection of the weighted residuals. Measurements were carried out under the same conditions (acquisition time and excitation intensity) to allow a quantitative and qualitative comparison between different experiments (pristine or exposed LB films).

### 4.4. Atomic Force Microscopy

Atomic force microscopy (AFM) height maps were obtained using an Anton Paar Tosca 400 afm, running in tapping mode with 70 kHz NuNano probes. Images were recorded at 1 Hz with a minimum resolution of 256 lines per 2 × 2 μm square. LB films of **1** were constructed with a 4-layer deposition and subsequently exposed to target analytes.

## 5. Conclusions

This work shows that fluorescent LB films can be constructed from amphiphilic DADQs. The average fluorescence lifetime can be correlated to the LB film structure and depends on packing (free volume) in the LB film and the compressibility of the LB films. Modulation of the fluorescence can be observed for exposure to the LB film by target analytes. Further work should be focused on developing DADQs for improved specificity in sensing organic and/or inorganic materials. Further studies should also establish the lowest possible detection limit for fluorescent LB films based on DADQ adducts. If successful, these fluorescent LB films could be developed in systems for environmental sensing.

## Figures and Tables

**Figure 1 molecules-27-03893-f001:**
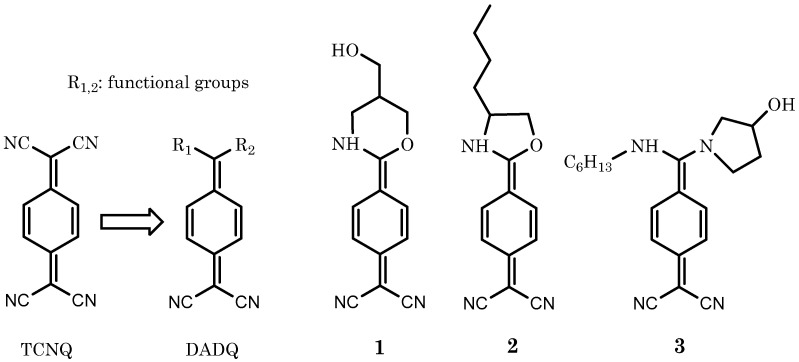
**Left,** chemical structure of un-functionalized TCNQ. The insertion of functional groups R_1_ (and R_2_) leads to the formation DADQs. To the right, the three DADQs used in this study. (**1**) (2-(4-(5-(hydroxymethyl)-1,3-oxazinan-2-ylidene) cyclohexa-2,5dien-1-ylidene)malononitrile), (**2**) (2-(4-(4-butyloxazolidin-2-ylidene)cyclohexa-2,5-dien-1ylidene)malononitrile) and (**3**) (2-(4-((hexylamino)(3-hydroxypyrrolidin-1-yl)methylene)cyclohexa2,5-dien-1-ylidene)malononitrile). See text for detail.

**Figure 2 molecules-27-03893-f002:**
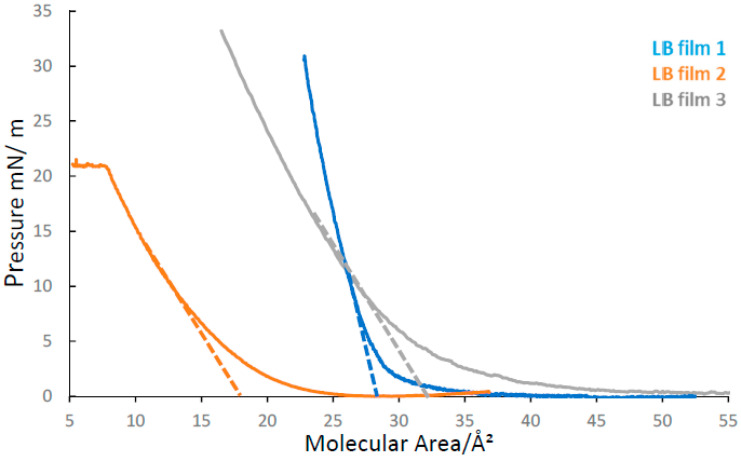
Isotherms for **1**, **2** and **3**. Molecular head areas shown in Table 1. The dashed line indicates a fit of the solid phase data to obtain the molecular head area. See text for detail.

**Figure 3 molecules-27-03893-f003:**
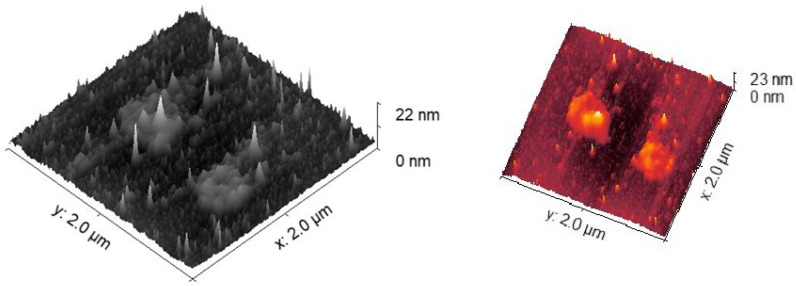
AFM height maps of pristine LB films of **1**. The number of layers is 4. The two images are different representations of the same AFM area scan. See text for details.

**Figure 4 molecules-27-03893-f004:**
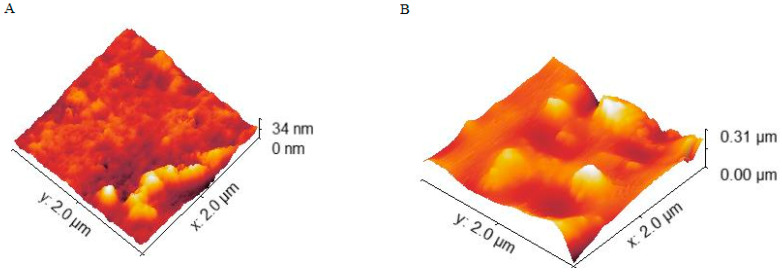
AFM height maps of LB films of 1 exposed to humic acid (HA); image (**A**), and human serum albumin (HSA), image (**B**). The number of layers is 4. See text for details.

**Figure 5 molecules-27-03893-f005:**
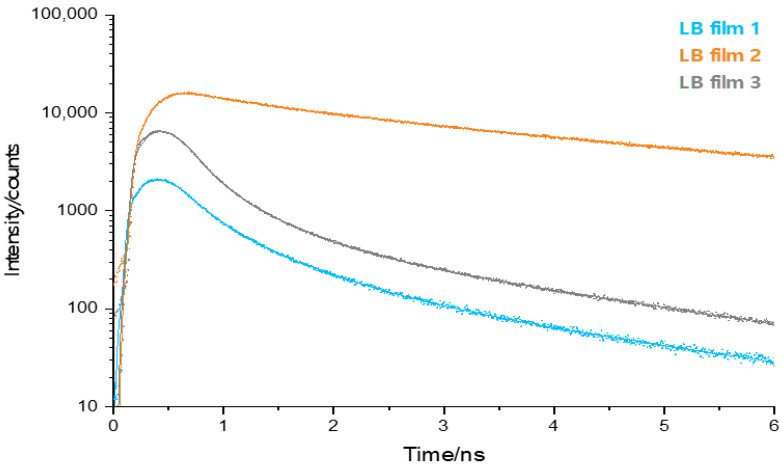
Time-resolved fluorescence microscopy decays for pristine (blank) LB films of **1**, **2** and **3**. Symbols are raw data and solid line is the fit. An IRF was used in the fitting procedure of the decays but not shown here. The fluorescence was detected using a long-pass filter (λ_em_ > 420 nm). The average lifetime data calculated from these decays can also be seen in Figure 6 (blank data set). For a breakdown of the individual decays into a sum of exponential terms, data can be found in the SI (Appendix A).

**Figure 6 molecules-27-03893-f006:**
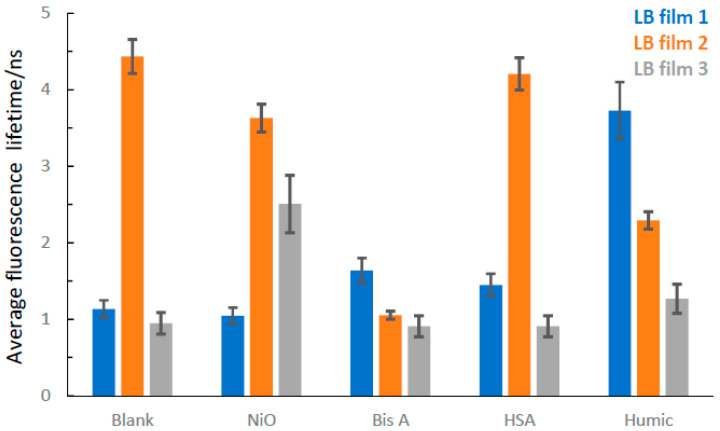
Average lifetimes for LB films of **1**, **2** and **3,** as calculated using Equation (4), with and without target analyte deposition. Blank is pristine LB film and humic is HA. See text for details.

**Figure 7 molecules-27-03893-f007:**
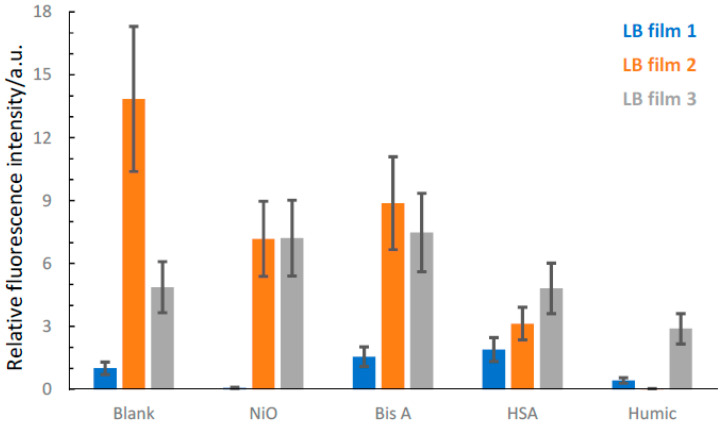
Relative intensities for LB films of **1**, **2** and **3** with and without target analyte deposition. Blank is pristine LB films and humic is HA. The relative intensities shown here have been calculated using Equation (5) and normalized to the intensity of pristine LB film **1**. The fluorescence was detected using a long-pass filter (λ_em_ > 420 nm). See text for details.

**Table 1 molecules-27-03893-t001:** Data obtained from the analysis of the LB isotherm experiments and parameters used in the LB film deposition. Transfer ratios were calculated for the 2nd layer. See text for details.

DADQ LB Film	Target Surface Pressure/mN × m^−1^	Molecular Head Area /Å^2^	Transfer Ratio /a.u.	Compressibility /mN × m^−1^
**1**	30	28.0 ± 3.0	1.03 ± 0.04	7.14 ± 0.82
**2**	18	18.0 ± 3.0	1.23 ± 0.04	38.15 ± 6.48
**3**	34	32.0 ± 4.0	0.97 ± 0.05	15.62 ± 2.02

**Table 2 molecules-27-03893-t002:** A summary of observations and conclusion for target analyte exposure to pristine LB films. The input is drawn from Figure 6 and Figure 7. Consistency refers to modulation of average fluorescence lifetime and integrated intensity being congruent. See text for details.

DADQ LB Film	1	2	3
NiO	No change in average lifetime. Decreased intensity. Anomaly?	Shorter average lifetime. Decreased intensity. Consistent observations.	Longer average lifetime. Decreased intensity. An anomaly.
Bisphenol A (Bis A)	Longer average lifetime. Intensity increase. Consistent observations	Shorter average lifetime. Decreased intensity. Consistent observations.	No change in average lifetime. Decreased intensity. Anomaly?
Humic acid (HA)	Longer average lifetime. Decreased intensity. An anomaly.	Shorter average lifetime. Decreased intensity. Consistent observations.	Longer average lifetime. Decreased intensity. An anomaly.
Human serum Albumin (HSA)	Longer average lifetime. Intensity increase. Consistent observations	No change in average lifetime. Decreased intensity. Anomaly?	No change in average lifetime. Decreased intensity. Anomaly?

## Data Availability

Not applicable.

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
