# Peer review of "Modulated Fluorescence in LB Films Based on DADQs—A Potential Sensing Surface?"

_molecules, 2022, doi:10.3390/molecules27123893_

Round 1

Reviewer 1 Report

Novel fluorescent Langmuir-Blodgett films were designed using varying functional groups in dicynaoquinodimethanes (DADQs). The effect of these varying functional groups on fluorescent behaviour with life time measurements are reported. The work is sufficiently novel and can be considered for publication in the molecules. However, the following minor modifications must be done before acceptance. 

  1. The lifetime graphs must be improved. The fitting of the graphs must be neatly presented with all the details of the fit used.
  2. Figure 7 shows the PL intensity variation bar graph. But authors are encouraged to present the FL original graph along with the variation bar graph. 
  3. The characterization of the films must be improved. At present except for AFM, no other characterizations of films are presented. 

Author Response

We would like to thank the referee for their positive and constructive comments to our work. The referee will see that the manuscript is revised significantly, also in response to the other referees. The changes in the manuscript are highlighted in red.

With regard to specific comment to the referee, we have the following responses; 

  1. The lifetime graphs must be improved. The fitting of the graphs must be neatly presented with all the details of the fit used. We have added clarification and explanations in the text but mainly in the caption to the figure. We make clear refence to another figure in the manuscript and we also refer to the SI and data tables for a breakdown of the decays into the sum of exponentials. The data in figure 5 does show raw data and fit (as stated in the caption but we decided to omit the IRF as with it included the figure becomes very messy and hard to evaluate. 
  2. Figure 7 shows the PL intensity variation bar graph. But authors are encouraged to present the FL original graph along with the variation bar graph. We elected not to use a standard fluorescence spectrometer and normal spectral measurements for this comparison. We hope that we have been able to explain the rational for doing this differently now in the revised version of the manuscript. We point out that spectra of the three LB films can be found in the SI, for comparison. The reason was that only in the microscope setup, it was possible to eliminate factors related to sample geometry and variation in excitation and detection conditions. Furthermore, the detection systems in the microscope is for various reasons far superior to that of a standard fluorescence spectrometer which isn't really suited for mono/bilayer LB films. For these reasons we elected to proceed with the integration of recorded fluorescence decays instead, which by definition will be proportional to a steady state spectra. With the filter used we can state that we collected the whole spectrum from the LB films.
  3. The characterization of the films must be improved. At present except for AFM, no other characterizations of films are presented. Unfortutaly we are limited in experimental resource to what more we can do and there is also the time aspect, to get such characterisation done will take time and no doubt have a significant impact on finalising this manuscript. We have tried other methods beyond AFM and this included FTIR which unfortutaly doesn't result in anything that can be published. We have also tried absorption but these films are extremely thin and therefore difficult to assess. We have attached a 4 layer LB film of compound 1 as an example of an exception although there are problems with this spectrum too, for 2-layer LB films the absorption isn't presentable. What we have done is to provide some more data from the AFM characterisation; cross-sectional height profiles for one target analyte deposition and this can now be found in the revised version of the SI. 

Reviewer 2 Report

molecules-1692034 – Sensing surfaces based on functionalised fluorescent dicynaoquinomethane Langmuir-Blodgett films.
1-    The introduction could be improved by referring to the following refs including 
https://doi.org/10.1021/acs.langmuir.9b03601
https://doi.org/10.1007/s00216-021-03660-6
 https://doi.org/10.1021/acs.langmuir.0c00045
https://doi.org/10.1039/C7CP04235C
which can offer help to compose a better writing for the introduction and retrieve useful information regarding this work as well.
2-    Analyses of data/results and focus of aims should be further elevated.
I recommend this work for publication after major revision.

Author Response

We would like to thank the referee for their positive and constructive comments to our work. The referee will see that the manuscript is revised significantly, also in response to the other referees. The changes in the manuscript are highlighted in red.

With regard to specific comment to the referee, we have the following responses; 

  1. We have taken the suggestion of the referee and included the references suggested. There is yet one more reference added to this work on the suggestion by another referee.
  2. We have added to the significantly to both the results section and to the discussion section as the referee will see.  These  additions include a further analysis of the Langmuir-Blodgett film data and more details to the optical measurements. This is also backed up with some additional data in the SI.

Reviewer 3 Report

This manuscript presents the synthesis of three DADQs, fabrication of their LB films and the use of these ultrathin films in optical (fluorescence based) sensing of some analytes.  While the overall work is interesting, there are several technical issues which need to be taken care of, before the manuscript can be considered for publication.  Some of the more prominent ones are listed below.  It is also to be noted that are several typos in the manuscript, which need to be corrected (eg. “.. could be used a range of …”, “… is the fore likely …”, “ … deposiation …”, “… targte analyet…” etc.).

  1. In many instances of DADQs, electronic excitation can lead to increase in the dihedral angle, as shown by published computational studies. This point should be noted.
  2. Discussions of the pi-A isotherms and extrapolated molecular areas in Fig. 2 are unclear. First, the structures of 1, 2 and 3 do not have the normal amphiphilic balance in the structure, with the hydrophobic part being low or minimal; so how does one explain the formation of the monolayer in each case?  2 seems to be particularly poor as it has a low molecular area as well as collapse pressure.  The inverse compressibility is a better measure of the strength and packing of the monolayer and should be discussed.  The correlation of the inferred molecular areas with the molecular structure is too qualitative.  It is preferable to make a simple computation of the molecular volume (using appropriate quantum chemical programs and optimized molecular structures) and assuming a reasonable length, estimate the cross-sectional areas that would be relevant to understand the extrapolated values from the pi-A isotherm.
  3. It is not clear if the TR given in Table 1 is for only the first layer. Since multilayer films are discussed later, the TR for multiple dipping should be provided to evaluate how well the multilayer structure is formed.
  4. The AFM images in Fig. 3 and 4 should be discussed in the light of line profiles drawn across the relevant morphological features with a discussion of the observed height estimates. In the absence of such information, if one were to assume that the feature shown in Fig. 3 is approximately 23 nm, it implies that each layer is approximately 5.75 nm or 57.5 angstrom.  How does one justify this on the basis of the length of the molecule, which should correlate with the layer thickness, if one assumes an edge-on orientation at the air-water interface?
  5. If possibilities like HA oxidizing the LB film surface exists, then the various speculations and interpretations of the observed sensing responses are questionable.
  6. The fluorescence responses are interpreted on the basis of the tightness of packing. The molecular area is used as the criterion of the tightness of packing.  This can be misleading, as the molecular area will critically depend on the orientation of the head group at the air-water interface.  This could be quite different between 1 - 3.  A better measure of the tightness of packing is the inverse compressibility (as noted above).  The authors should look at this aspect; it is not clear if the fluorescence response has a direct correlation with this more realistic measure of the packing efficiency.  Hence the interpretation of the observed optical responses is unclear. 
  7. 4.1: The purity of reagents and analytes must be clarified. Characterization of the new compounds should be provided, with at least elemental analysis, nmr / mass and infrared spectroscopy data on each compound.  In the supporting information, 3-pyrrolidinol is mispelt.
  8. The experimental method mentions that solutions of the analytes were deposited on the LB film, and subsequently dried. How does one rule out the possibility that the LB film is affected physically (other than electronic interaction with the analyte)?  For example, the film could dissolve locally and redeposit where the aqueous solution shrinks to, while evaporating. At the micrometric or nanometric level, these are distinct possibilities and any optical measurements on the ultrathin film can be seriously affected by such changes.  In fact, the non-uniformity of fluorescence responses mentioned in the manuscript could be resulting from such issues.  Attributing the consequences of such effects, to the electronic interaction would be misleading.
  9. The purity of water used as the subphase should be given in terms of resitivity, not resistance.
  10. Since the title itself specifies the sensing application, it is important to present the sensing responses as a function of the analyte concentration, and check for linearity of response, limit of detection etc., at least as a preliminary demonstration of the application possibility.

Author Response

We would like to thank the referee for their positive and constructive comments to our work. The referee will see that the manuscript is revised significantly, also in response to the other referees. The changes in the manuscript are highlighted in red.

With regard to specific comment to the referee, we have the following responses; 

This manuscript presents the synthesis of three DADQs, fabrication of their LB films and the use of these ultrathin films in optical (fluorescence based) sensing of some analytes.  While the overall work is interesting, there are several technical issues which need to be taken care of, before the manuscript can be considered for publication.  Some of the more prominent ones are listed below.  It is also to be noted that are several typos in the manuscript, which need to be corrected (eg. “.. could be used a range of …”, “… is the fore likely …”, “ … deposiation …”, “… targte analyet…” etc.).

We apologise for this, too late did we not realise that the spelling checks where not working. We have gone through the manuscript and (hopefully) dealt with all spelling mistakes now. 

  1. In many instances of DADQs, electronic excitation can lead to increase in the dihedral angle, as shown by published computational studies. This point should be noted.                                                                                          We have added a new comment in the manuscript, highlighting this possible scenario. We have also added a reference that we think is appropriate in this context.
  2. Discussions of the pi-A isotherms and extrapolated molecular areas in Fig. 2 are unclear. First, the structures of 1, 2 and 3 do not have the normal amphiphilic balance in the structure, with the hydrophobic part being low or minimal; so how does one explain the formation of the monolayer in each case?  2 seems to be particularly poor as it has a low molecular area as well as collapse pressure.  The inverse compressibility is a better measure of the strength and packing of the monolayer and should be discussed.  The correlation of the inferred molecular areas with the molecular structure is too qualitative.  It is preferable to make a simple computation of the molecular volume (using appropriate quantum chemical programs and optimized molecular structures) and assuming a reasonable length, estimate the cross-sectional areas that would be relevant to understand the extrapolated values from the pi-A isotherm.                                                We have provided new comments on the aspect of amphiphilic balance in the three systems DADQ systems used in this work. It is likely that the low amphiphilic character of 2 (as pointed out by the referee) has resulted in this system going into the bulk rather than remaining on the surface. We can rationalise this hypothesis from the obtained unusual transfer ratio for LB film deposition for 2, we now acknowledge this in our revised version. We emphasize though, that we are confident we have an LB film of 2, the optical data clearly supports this. We have taken advice of the referee to analyse the compressibility of the LB films, to us it made more sense to produce this quantity rather than its inverse, and the data for each LB film is listed in a new column in Table 1. These new results are discussed in the discussion section as well. Regarding the suggestion by the referee to perform quantum chemical calculations to obtain estimates of molecular volumes, we regret to say that this has not been possible to achieve within the time frame given to revise this manuscript. We sought to mobilise some expertise in this area (as we are novices in computational chemistry) with the aim to deliver something of a quality desired but it is clear that this would take much more time than what we have available here to re-submit a revised manuscript.   
  3. It is not clear if the TR given in Table 1 is for only the first layer. Since multilayer films are discussed later, the TR for multiple dipping should be provided to evaluate how well the multilayer structure is formed.                We confirm that it is for a 2 layer film. We have added this information in the caption to the Table 1 title. 
  4. The AFM images in Fig. 3 and 4 should be discussed in the light of line profiles drawn across the relevant morphological features with a discussion of the observed height estimates. In the absence of such information, if one were to assume that the feature shown in Fig. 3 is approximately 23 nm, it implies that each layer is approximately 5.75 nm or 57.5 angstrom.  How does one justify this on the basis of the length of the molecule, which should correlate with the layer thickness, if one assumes an edge-on orientation at the air-water interface?                                                                                We have provided additional comments to the aspects of molecular lengths and also made an estimation of molecular length that are reasonable and/or expected. We would also like to point out here (and we do so in the manuscript as well) that the height bar to the right of the AFM maps appears to have been scaled by the spikes in the AFM map. These spikes are likely due to the underlying roughness of the glass slide and not a reflection of the LB film thickness. In addition, we have provided some additional data showing the cross section of the LB film with (and without) HA deposition. This can be found in the revised version of the SI. 
  5. If possibilities like HA oxidizing the LB film surface exists, then the various speculations and interpretations of the observed sensing responses are questionable.                                                                                                    We acknowledge that this is a possible in some cases but we do not believe that this is a general scenario due to target analyte deposition in this work. For the case of HA deposition we put forward the hypothesis of oxidation of the LB film as HA has acid character. With oxidation of the LB film we would expect breakdown of the LB film into smaller less conjugated organic materials. Such material would not absorb light at 400 nm (the laser excitation wavelength) which explain why we observe an fluorescence intensity decrease. However, we also observe fluorescence intensity increase (and fluorescence lifetime increase) for certain target analyte exposures and this is inconsistent with a picture of breakdown of organic materials into smaller less conjugated systems. Therefore we believe the LB film to be intact in those cases and the fluorescence (lifetime) increase is a reflection less non-radiative decay of TCNQ adduct due to target analyte interactions.   
  6. The fluorescence responses are interpreted on the basis of the tightness of packing. The molecular area is used as the criterion of the tightness of packing.  This can be misleading, as the molecular area will critically depend on the orientation of the head group at the air-water interface.  This could be quite different between 1 - 3.  A better measure of the tightness of packing is the inverse compressibility (as noted above).  The authors should look at this aspect; it is not clear if the fluorescence response has a direct correlation with this more realistic measure of the packing efficiency.  Hence the interpretation of the observed optical responses is unclear.                    We have partly answered this in #2, we have now calculated compressibilities for each LB film and the data is presented in Table 1 in a new column. The outcome of this analysis is that the LB film of 2 stands out as it shows the highest compressibility which we interpret as the highest packing in the film of the 3 systems studied. This is then reflected in the fluorescence decays as the lifetime measured is the longest for this system, which is due to small free volume and limited non-radiative decay. We have added to the manuscript aspects of this in the result section as well as in the discussion section.
  7. 4.1: The purity of reagents and analytes must be clarified. Characterization of the new compounds should be provided, with at least elemental analysis, nmr / mass and infrared spectroscopy data on each compound.  In the supporting information, 3-pyrrolidinol is misspelt.                                      We have provided this information and is added to the SI. Note that we did not perform any additional chemical manipulation and/or modifications of the selected target analytes, this is stated in the method and materials section of the manuscript.
  8. The experimental method mentions that solutions of the analytes were deposited on the LB film, and subsequently dried. How does one rule out the possibility that the LB film is affected physically (other than electronic interaction with the analyte)?  For example, the film could dissolve locally and redeposit where the aqueous solution shrinks to, while evaporating. At the micrometric or nanometric level, these are distinct possibilities and any optical measurements on the ultrathin film can be seriously affected by such changes.  In fact, the non-uniformity of fluorescence responses mentioned in the manuscript could be resulting from such issues.  Attributing the consequences of such effects, to the electronic interaction would be misleading.                                                                                                  The referee is correct that these are in principle possible scenarios and we have provided additional data from control experiments to demonstrate the effect of a dissolved LB film on the fluorescence kinetics (data shown in the revised version of the SI). This data is obtained (from tests on all three systems) using a long chain alcohol (pentanol) as the target analyte. The results of this exposure is a fast component (limited by the time-resolution of the detection system) on time-scale of 30-50 ps, also with a considerable amplitude (yield). Our interpretation is that the LB film has indeed been dissolved in this case, and that the fast fluorescence kinetics is a reflection of DADQ in a solution phase where the fluorescence is limited by the conformational change. We note that we do not observe such a fast fluorescence component with such a high amplitude (yield) in the other exposure data shown in the main manuscript. The possibility of a dissolved LB film left to dry before measurements would also lead to a the appearance of a fast decays phase in line with our our experience from earlier work on DADQs. This is because the DADQs have a very large ground state dipole moment and this would in turn lead to head-to-tail aggregates with strong intermolecular interactions, and very likely a low(ered) fluorescence intensity. We have added to the text in the result section of the manuscript about this control experiment and its outcome.
  9. The purity of water used as the subphase should be given in terms of resitivity, not resistance.                                                                                    We have made this correction, see the Material and method section of the main manuscript. 
  10. Since the title itself specifies the sensing application, it is important to present the sensing responses as a function of the analyte concentration, and check for linearity of response, limit of detection etc., at least as a preliminary demonstration of the application possibility.                            The referee makes a valid point here and we have in this present work not focused on detection limits, instead the focus has been on the detection capabilities of a smaller variety of different chemical systems. We have added some comments in the discussion section, on the basis of area coverage estimated in the AFM experiment, about % area exposure that would be required to detect a fluorescence modulation (we give a conservative estimate). But in order to address the comment by the referee we have tuned down the language about these LB films being sensing platforms (at this stage). Instead we focus on detection capabilities (of different types of chemical systems) and limit the discussion to suggest that these LB films could be developed into future sensing platforms (see conclusion section). Furthermore, to avoid any confusion we have altered the title of the work in order to address the comment made by the referee.

Round 2

Reviewer 2 Report

After this revision it in the manuscript ID "molecules-1692034" this paper is accept in present form and can be a very good contribution to Molecules.

Reviewer 3 Report

The authors have addressed all the issues raised in the earlier report.  Most of the responses and revisions carried out are fine.  The manuscript can be accepted for publication.  However, the authors should check through the manuscript once again very carefully, as there are several typos still seen.  A few examples:  Page 3, Line 107, Page 6, Line 220, Page 9, Line 339, Page 12, Line 456  (unit of resistivity is wrong).

It is also seen that the authors have uploaded a draft SI with internal discussion comments and several data still not included, or proper formatting not done.  The SI should be updated with all the data, and the error-free and complete final version (without comments) should be uploaded.